# Effect of Shock Waves on the Growth of *Aspergillus niger* Conidia: Evaluation of Germination and Preliminary Study on Gene Expression

**DOI:** 10.3390/jof8111117

**Published:** 2022-10-24

**Authors:** Daniel Larrañaga-Ordaz, Miguel A. Martínez-Maldonado, Blanca E. Millán-Chiu, Francisco Fernández, Eduardo Castaño-Tostado, Miguel Ángel Gómez-Lim, Achim M. Loske

**Affiliations:** 1Posgrado en Ciencia e Ingeniería de Materiales, Centro de Física Aplicada y Tecnología Avanzada, Universidad Nacional Autónoma de México, Blvd. Juriquilla 3001, Querétaro 76230, Mexico; 2Centro de Física Aplicada y Tecnología Avanzada, Universidad Nacional Autónoma de México, Blvd. Juriquilla 3001, Querétaro 76230, Mexico; 3CONACyT—Centro de Física Aplicada y Tecnología Avanzada, Universidad Nacional Autónoma de México, Blvd. Juriquilla 3001, Querétaro 76230, Mexico; 4Facultad de Química, Universidad Autónoma de Querétaro, C.U., Cerro de las Campanas s/n, Querétaro 76010, Mexico; 5Centro de Investigación y Estudios Avanzados del Instituto Politécnico Nacional, Unidad Irapuato, Km 9.6 Libramiento Norte Carretera Irapuato-León, Irapuato 36824, Mexico

**Keywords:** shock waves, acoustic cavitation, gene expression, *Aspergillus niger*, cell permeabilization, fungal germination

## Abstract

Shock waves, as used in medicine, can induce cell permeabilization, genetically transforming filamentous fungi; however, little is known on the interaction of shock waves with the cell wall. Because of this, the selection of parameters has been empirical. We studied the influence of shock waves on the germination of *Aspergillus niger*, to understand their effect on the modulation of four genes related to the growth of conidia. Parameters were varied in the range reported in protocols for genetic transformation. Vials containing conidia in suspension were exposed to either 50, 100 or 200 single-pulse or tandem shock waves, with different peak pressures (approximately 42, 66 and 83 MPa). In the tandem mode, three delays were tested. To equalize the total energy, the number of tandem “events” was halved compared to the number of single-pulse shock waves. Our results demonstrate that shock waves do not generate severe cellular effects on the viability and germination of *A. niger* conidia. Nevertheless, increase in the aggressiveness of the treatment induced a modification in four tested genes. Scanning electron microscopy revealed significant changes to the cell wall of the conidia. Under optimized conditions, shock waves could be used for several biotechnological applications, surpassing conventional techniques.

## 1. Introduction

*Aspergillus niger* is one of the most important fungi used to produce food ingredients, pharmaceuticals and industrial enzymes [1]. An example is the production of over 1.75 million tons of citric acid annually [2,3,4,5]. Its proliferation occurs through nonmotile spores, that is, via single-nucleus cells called conidia. Modifying the structure of their cell wall has potential industrial relevance [6,7,8,9,10,11] because it may enhance the response to stress [6] and increase the production of compounds and metabolites [12,13]. 

The germination stage of conidium is preceded by a process during which it passes from a vegetative stage to an active state [12,14]. Establishing the conditions of this process is important to optimize the use of *A. niger* to obtain compounds of industrial relevance [15]. It has been suggested that the use of emerging technologies could promote the growth of fungal cells or the production of enzymes from bacteria and fungi [16,17,18,19]. Disturbing the surface of the cell wall may also increase the efficiency of genetic transformation, which is important to produce large amounts of biomass and a wide variety of metabolites, enzymes and compounds, such as antibiotics, insulin, hepatitis vaccines and anticoagulants [20,21,22]. Unfortunately, standard genetic transformation methods are cumbersome, and have low efficiency and poor reproducibility.

Underwater shock-wave-induced transformation of fungi has proven to be an option to solve these difficulties [23,24,25,26,27,28]. During the transformation of filamentous fungi, acoustic cavitation is believed to be the main phenomenon acting on the cell walls [25,29,30]; however, little is known about the details of the mechanisms involved.

Shock waves used for the transformation of bacteria and fungi have a 0.5–3 μs compression peak (*p^+^* = 10–150 MPa), followed by a 2–20 μs rarefaction pulse (absolute *p^−^* value ~5–30 MPa). The bubble dynamics of shock-wave-exposed fluids are influenced by the magnitude of the peak positive and negative pressure pulse, the full width at half maximum (FWHM) and the rise time (*t_r_*) of the shock wave [31,32,33,34]. These parameters depend on the generation principle, focusing mechanism, initial energy, type of vial, water conditions, and suspension inside the vial. Acoustic cavitation, formed from microbubbles and nucleation sites contained in a shock-wave-exposed fluid, produces shear stress, secondary shock waves, and high-speed microjets affecting conidia in the nearby vicinity. As the positive pressure pulse arrives, these tiny bubbles are compressed. After the shock wave passes, the pressure difference between the interior and exterior of each bubble and the trailing tensile phase of the pressure profile, trigger bubble growth. Between approximately 50 and 700 µs later, the bubbles suffer a violent asymmetric collapse. Because of this, the fluid on one side of the bubble accelerates inwardly faster, resulting in the emission of a secondary shock wave and the development of a high-speed (up to 700 m/s) microjet of fluid. These microjets burrow through the bubble acting as microsyringes capable of injecting fluid from the outside into the cell [31,33,35]. They have been used to genetically transform *A. niger* [28,29]. Microbubble collapse and shear stress also have been reported to be the main mechanisms producing effects in human cells [36]. 

The microjet emission can be enhanced if a second shock wave arrives during bubble collapse [37,38]; however, if the second shock wave arrives too early, it may suppress the bubbles generated by the first shock, reducing the bubble collapse energy. The influence of these so-called tandem shock waves on membrane permeabilization was confirmed by showing that they significantly improved DNA delivery into microscopic fungi [29].

To a certain extent, transformation efficiency depends on the diameter of the microjets. Severe damage may occur to the cells if their size is too large. The diameter of microjets emitted by bubbles that suffered a violent asymmetric collapse is approximately one tenth of the bubble diameter [31,39,40,41]. Unfortunately, it is not possible to control the size distribution of the bubbles contained in a conidial suspension. Nevertheless, tandem shock waves generated at a specific delay selectively enhance the collapse energy of bubbles having certain diameter.

*Aspergillus niger* can reproduce both sexually and asexually. Its asexual lifecycle starts with one conidium. The conidium has an average size between 3 and 5 µm. Its hydrophobic wall allows it to be transported by the wind while remaining dormant, until it finds an environment with enough nutrients to activate its metabolism [42]. From this moment on, the conidium starts so-called isotropic growth [43]. After approximately 6–8 h, it develops into a germ tube [44]. Later, the germ tube grows by apical extension, forming a multinucleated cell divided by septa, referred to as hypha. Some ramifications develop conidiophores, that is, structures that produce clones of the original conidium [10,12,45]. 

The development of conidial fungi begins with the germination process, which is classified into two growth stages: isotropic and polarized [44,46]. Isotropic growth occurs during the first morphological change in germination. It includes water uptake and growth due to the addition of new cell wall material [12] via the activation of metabolic activities (DNA, RNA, and protein synthesis). At the end of this stage, the cell diameter increases twofold or more. During growth, the swollen conidia deform, leading to a germ tube. The resulting tube is characteristic of polarized growth [44]. The final stages are characterized by an increase in the speed of growth of the germ tube [14], triggering the functional organization of the hyphal tip, which will establish some interhyphal fusions by branching creating a fungal mycelium [47].

The cell wall of filamentous fungi is an essential component for morphogenesis and cell viability as well as protection against external ambient stress factors [48,49,50]. Its structure is a physically rigid layer formed mainly of networks of carbohydrates (β-1,3 glucan, β-1,6 glucan, and chitin), proteins (mannoproteins and hydrophobins), and other surface components, such as melanins [7,51]. 

Modification of the fungal cell wall can be induced either by changing its physical properties or by triggering molecular signaling pathways [52]. Physical treatments such as heating and exposure to microwaves, ultrasound, shock waves and other pressure variations can affect its shape and structure. As reported by Gomez-Gomez et al. [53], the application of ultrasound on fungal spores resulted in cell wall wear, reflected by a thinner wall, an uneven width, and the appearance of some dissolved areas. These modifications can activate the pathway of cell wall integrity (CWI) induced by the response of fungal conidia to environmental stimuli [7]. The signaling of the CWI is induced by complex genetic machinery and has an impact on wall remodeling. One mechanism of this pathway is the activation of the putative protein sensors (Wsc1p-Wsc4p) cell-wall-stress responsive components. They stimulate the small Ras homologous (Rho) GTPases, which are signaling molecules that control several downstream mechanisms, ensuring cell wall biogenesis, actin organization and polarized secretion. Six Rho GTPases have been identified in *A. niger* where *RhoA* seems to have a central role in polarity and cell viability, while *RhoC* apparently has less influence on conidial growth [9,54,55].

The main goal of this research was to study the influence of underwater shock waves on the germination of *A. niger*. Understanding their effect on the modulation of the genes *β-Actin*, *Wsc4*, *RhoA* and *RhoC*, which are related to the stress pathways and are involved in the development of conidia was a priority. Furthermore, the experiments were designed to focus on the determination of parameters such as the peak positive pressure, FWHM, delay of tandem shock waves, and water conditions to diminish conidia mortality. This could contribute to the optimization of shock wave applications for genetic transformation, and result in a better production of metabolites and compounds of industrial relevance, while maintaining relatively high cell viability. Our results can provide useful information to the biotech industry and for groups doing research on shock wave exposure of *A. niger*. To the best of our knowledge, this is the first study of the effects of shock waves on conidial germination.

## 2. Materials and Methods

### 2.1. Experimental Setup

As shown in Figure 1, the experimental shock wave generator consists of a Lucite water tank (base 675 × 675 mm, height 450 mm) with a precision *xyz* positioning system UniSlide Assemblies (Velmex, Inc., Bloomfield, NY, USA) mounted on top of it. A piezoelectric shock wave source (Piezolith 2501, Richard Wolf GmbH, Knittlingen, Germany) was used to generate either single-pulse or tandem shock waves by high-voltage discharges applied to an array of 3000 piezoelectric crystals mounted on a hemispherical bowl-shaped backing. Approximately 230 µs after the electric discharge, a shock wave arrives at the center of the sphere (*F*). The electroacoustic transducer is self-focusing because of the spherical shape of the arrangement. The nonlinearities and steepening of the pressure pulse during propagation produces a shock wave at the focal zone. The concave side of the shock wave source at the bottom of the tank is insulated by a polymer. To be able to generate tandem shock waves at the desired delay, a pulse generator triggers two discharge circuits, that is, two capacitors are charged by a high voltage transformer until their trigger is fired. With this system, two shock waves can be emitted with an adjustable delay between 50 and 950 μs. The experimental setup was performed according to what is described in a previous publication [29].

### 2.2. Pressure Measurements and High-Speed Imaging

A total of five pressure profiles were recorded inside one of the water-filled vials described below, using a polyvinylidene difluoride (PVDF) needle hydrophone (Imotec GmbH, Würselen, Germany) with a 20 ns rise time and fed into a 300 MHZ digital oscilloscope (Tektronix Inc., Beaverton, OR, USA, model TDS3032). The water level and temperature were fixed at 80 mm above *F*, and 25 °C, respectively. The tank was filled with tap water for both pressure measurements and shock wave exposure to the suspension-filled vials. 

As an aid to select convenient delays for the tandem mode, a high-speed Motion Pro x4 (Integrated Design Tools, Inc., Pasadena, CA, USA) camera was used to record cavitation bubbles inside a sealed vial containing water and air. Images of bubble expansion and evolution were captured at 30,000 frames per second (fps) in the single-pulse mode, using a discharge voltage of 4.0, 5.0 and 6.0 kV. All reported voltages had an uncertainty of ±0.125 kV.

### 2.3. Fungal Cell Culture and Sample Preparation

*Aspergillus niger* strain CBS 513.88 was grown for 6 days on *Aspergillus* minimal medium agar to promote conidiation at 30 °C as described by Kaminskyj [57]. Conidia were harvested from the sporulated fungus by adding 5 mL of *Aspergillus* minimal medium broth. To separate the mycelium and conidial heads, the suspension was filtered with 3 layers of Miracloth (EMD Millipore, MA, USA, catalog number: 475855). Next, the suspension was diluted to a concentration of 1.5 × 10^6^ cells per milliliter. The conidia were counted with a Neubauer chamber. Conidia were chosen for this study because it is relatively easy to separate them.

### 2.4. Shock Wave Application

For each treatment, 1.5 mL of conidia in suspension was transferred into 4 mL cigar-shaped (41 mm-long, 13 mm in diameter) sterile polyethylene transfer pipettes (Thermo Fisher Scientific, Waltham, MA, USA) that were cut approximately 20 mm above the stem and heat-sealed before the shock wave treatment. Each vial was positioned vertically so that the center of the suspension coincided with the focus *F* of the shock wave generator, using two laser pointers mounted horizontally at the focal plane on perpendicular sides of the water tank (Figure 1). The error in positioning the vials was estimated to be ± 0.5 mm. Three peak positive pressure values (42.04 ± 1.06 MPa, generated at 4 kV; 66.48 ± 1.49 MPa, generated at 5 kV; and 82.77 ± 1.21 MPa, generated at 6 kV) and two shock wave modes (single-pulse and tandem) were tested. The peak positive pressure values and the number of applied shock waves were chosen based on previous reports [23,29] and preliminary experiments (data not shown). For simplicity, instead of mentioning the three pressure values used, in the remaining part of the article we will refer to their corresponding discharge voltages, i.e., 4, 5 and 6 kV. The number of shock waves applied was 50, 100 or 200. Fifty-seven sample vials were divided into 18 treatment groups and 1 control group. Each group consisted of three vials, independently exposed to the same conditions. All groups were randomized before shock wave exposure. The discharge rate was fixed at 0.5 Hz for both single-pulse and tandem shock waves. To equalize the total energy input to each vial, the number of tandem “events” (one event consisting of a pair of shock waves) was halved compared to the number of single-pulse shock waves. The water level and the water temperature were the same as for the pressure measurements.

### 2.5. Measurement of Cell Viability

Immediately after conidia exposure to shock waves, serial dilutions (1:10) were made for each sample until reaching 1.5 × 10^4^ conidia/mL. From this, 25 µL were inoculated onto Petri dishes with minimal medium agar. They were incubated at 30 °C for 48 h. Colony-forming units (CFU) were counted against the light. The calculation per milliliter was made using the formula (CFU = number of colonies × dilution)/inoculation volume. The development of the control group was compared with the shock-wave-exposed groups. All cell viability tests were performed in triplicate.

### 2.6. Morphological Analysis and Its Quantification

Petri dishes were covered with a film of minimal media agar, and 50 µL of conidia suspension from each vial was added to each plate and covered. The samples were observed under a BX40 optical light transmission microscope (Olympus Co., Tokyo, Japan) 0, 4, 6, 8, 10 and 12 h after shock wave or sham treatment at a storage temperature of 30 °C. A total of 100 conidia were analyzed per sample (vial). Photographs were taken with a CS2100M-USB camera (Thorlabs, Inc., Newton, NJ, USA). The analysis of the photographs was conducted with image processing software (ImageJ, National Institutes of Health, Bethesda, MD, USA). The first determination was the percentage of conidia that developed a germ tube between the sixth and the twelfth hour. The second observation was the swelling of the conidia by measuring their area from the beginning until the tenth hour. All measurements were performed by the same person.

### 2.7. Statistical Design and Data Analyses

Combining the voltage (4, 5 or 6 kV), the number of shock waves (50, 100 or 200) and the shock wave mode (single-pulse or tandem) resulted in 19 experimental groups, i.e., 18 (3 × 3 × 2) treated and 1 control group (without shock wave exposure), to be treated in a randomized order. Cell viability among treatments was statistically compared using a Kruskal–Wallis test. With respect to the morphological analyses, for each of the 19 experimental groups, the corresponding data were considered as time profiles of repeated measures along the sampling times mentioned before. To interpret differences between treatments, a local polynomial curve was adjusted for each group. Bootstrap intervals at 95% were obtained [58] using the R software environment, which is a free software for statistical computing and graphics that runs on a wide variety of UNIX platforms, Windows and Mac OS [59].

### 2.8. Scanning Electron Microscopy

Treated and control conidia fixation was performed by mixing one volume of conidial suspension and one volume of glutaraldehyde (Electron Microscopy Sciences, Hatfield, PA, USA), dissolving in sodium cacodylate buffer 0.1 M, pH 7.4 (Electron Microscopy Sciences) and incubating for one hour. After this, the samples were centrifuged at 7500× *g*. The supernatant was removed, and the pellets were washed two times with sodium cacodylate buffer 0.1 M, pH 7.4 for 10 min and stored for 24 h at 4 °C. After that, the sodium cacodylate buffer was replaced with osmium tetroxide (Electron Microscopy Sciences) in sodium cacodylate buffer, and the cells were kept there for 1 h. Then, the dehydration process was started by submerging the samples in ethanol (10, 30, 50, 70 and 100%) for 10 min in each solution. Finally, the ethanol was replaced by liquid CO_2_ with a critical point dryer CPD2 (Ted Pella, Inc., CA, USA). The conidia were coated with a gold nanoparticle layer (20 nm) using a sputter coater EMS 550 (Electron Microscopy Sciences) and observed with a JSM-6060LV (JEOL, Tokyo, Japan) scanning electron microscope.

### 2.9. Cell Permeabilization Test

Twenty-one vials containing 1 mL of a suspension of 1 × 10^6^ conidia/mL and 400 µM of fluorescein isocyanate-dextran 10 kDa (FD10-FITC) (Sigma Aldrich, St. Louis, MO, USA, catalog number FD10S) were divided into 3 groups of 3 vials and exposed to 200 singe-pulse shock waves, generated at either 4, 5 or 6 kV. Analogously, 3 groups of 3 vials were exposed to 100 tandem shock waves, generated at 4, 5 and 6 kV. A non-shock-wave-treated control group consisted of the same number of vials. After treatment, all samples were recovered, washed three times with 2 mL of PBS, centrifuged 5 min at 1500× *g*, resuspended in 4% paraformaldehyde, and fixed for 30 min. Three additional washes with PBS followed. Finally, all samples were saved in the dark at 4 °C. Previously to the observation, 20 µL of calcofluor white (1%) was added to an equal sample volume. Samples were analyzed using a confocal laser scanning microscope (LSM880, Carl Zeiss, Jena, Germany), using a 20× objective lens, and a 1.5 and 2 digital zoom (EC Plan-Neofluar-Zeiss). Green fluorescence was observed with excitation at 488 nm and emission at 505 nm, and blue fluorescence with excitation at 360 nm and emission at 420–465 nm. 

### 2.10. RNA Extraction

For RNA isolation, 1.5 mL of conidia was analyzed in duplicate. Briefly, the samples were centrifuged at 16,600× *g* at 5 °C for 3 min. Then, the supernatant was discarded and the pellet was washed with 500 µL RNAse-free water. Afterward, the pellets were frozen in liquid N_2_ and the RNA was extracted using an Allprep Fungal DNA/RNA/Protein kit (cat no. 47154, www.qiagen.com) as follows. The conidia were resuspended in 367.5 µL of the HC solution and 3.6 µL of B-mercaptoethanol and the solution was vortexed for 2 min. Then, the sample solution was transferred to a power bead tube (Qiagen GmbH, Hilden, Germany, catalog number 13116-50), frozen in liquid N_2_, and vortexed at a maximum speed for 5 min. This step is the lysis step and it was repeated 3 times. Next, 175 µL of the MR solution was added and thoroughly mixed for 2 min. The lysate was transferred to an MB spin column (Qiagen GmbH, catalog number 47154) and centrifuged for 2.5 min. After this, 350 µL of the RB solution was added to the flow-through sample and vortexed for 30 s. The lysate was transferred to a new MB spin column and centrifuged at 16,600× *g* for 2.5 min. After this, the MB spin column was placed in a clean 2 mL tube and 650 µL of cold RW solution was added. The column was centrifuged at 16,600× *g* for 3 min, 650 µL of cold ethanol was added and the column was centrifuged twice at the same speed for 3 min. Then, the spin column was placed in a clean 2 mL tube, adding 100 µL RNase-free water into the center of the filter membrane, and centrifuged at 16,600× *g* for 2.5 min for RNA collection. The RNA concentration and quality were determined using a Nanodrop 1000 spectrophotometer (Thermo Fisher Scientific, Waltham, MA, USA). The RNA integrity was assessed with 1% agarose gel electrophoresis in denaturing conditions at 100 V for 45 min. Finally, it was stained with SYBR safe gel for 5 min and visualized with a Gel Doc Ez Imager documentation system (C1000 touch thermal cycler, Bio-Rad Laboratories, Inc., Hercules, CA, USA).

### 2.11. RT-PCR

The retrotranscription of the RNA of the isolated samples into cDNA was performed by the reverse transcription–polymerase chain reaction according to the manufacturer’s manual (Taq DNA Polymerase Recombinant, Invitrogen by Thermo Fisher Scientific, Waltham, MA, USA). The PCR amplification was achieved using the primers described in Table 1. The reaction was conducted in a final volume of 10 µL containing 0.5 µM of each primer, 200 µM of each of the deoxyribonucleotide triphosphates (dNTPs), 1× Phusion^TM^ plus reaction buffer (Invitrogen), 0.02 U/µL of Taq DNA polymerase (Invitrogen), and ≈30 ng of cDNA. The end-point RT-PCR conditions were as follows: denaturation at 98 °C for 30 s; 30 cycles of 30 s at 98 °C, 30 s at 62 °C, and 60 s at 72 °C, followed by 5 min at 72 °C. The PCR was performed with the C1000 touch thermal cycler (Bio-Rad Laboratories, Inc. Hercules, CA, USA). After amplification, the PCR products (4 µL) were assessed by electrophoresis in 1.2% agarose gels at 100 V for 45 min, followed by staining with SYBR Safe DNA gel stain (Invitrogen) for 5 min. The stained gels were observed, and the intensity of the bands was analyzed by relative quantification using the software coupled to the documentation system Gel Doc Ez Imager (Bio-Rad Laboratories, Inc., Hercules, CA, USA) by comparison with the molecular weight marker (Invitrogen SM0331) as a standard.

## 3. Results

### 3.1. Pressure Measurements and High-Speed Imaging

According to the recorded pressure profiles, the transfer pipette did not noticeably change the shape of the waveform. Inside the water-filled test vial, the −6 dB focal volume, defined as the volume within which the positive pressure is at least 50% of its peak value, had the shape of an ellipsoid with a minor and major axis of approximately 2 and 13 mm, respectively. At discharge voltages of 4, 5, and 6 kV, the shock wave source generated a peak positive pressure (± absolute uncertainty) of 41.69 ± 1.06 MPa, 65.94 ± 1.49 MPa, and 82.10 ± 1.21 MPa, respectively. The corresponding FWHM values (±systemic uncertainty) for 4, 5, and 6 kV were 232, 196, and 200 ± 8 ns, respectively.

Figure 2 shows bubble expansion and evolution inside and around a vial containing 1.5 mL of water, exposed to a single-pulse shock wave generated at 4, 5, and 6 kV. The initial time (*t* = 0) was defined to correspond to the frame where the first bubbles could be observed. At *t* = *T*, the shock wave had already traversed the vial and was reflected off the water–air interface inside the vial, generating a large bubble cloud. This occurred because the acoustic impedance of water is higher than that of the air inside the vial, inverting the phase of the reflected shock wave and enhancing cavitation. A higher bubble density was also observed at the water–polyethylene–water interface. Clouds inside the vials were formed by bubbles, reaching sizes of up to approximately 2 mm. Bubbles invisible to the naked eye also were formed, slightly changing the image hue in some regions (see circles on the 4 kV frames). 

The criterion used to select an adequate delay in the tandem mode was to send the second shock wave at the time when bubbles having the size of the conidia (~5–10 µm) started to collapse, expecting to reinforce microjets with diameters of 1 µm or less. Our hypothesis was that cells can repair microjet-induced holes of approximately one tenth of their size. On our high-speed images, it was impossible to distinguish individual bubbles with a diameter less than 0.01 mm. Because of this, we focused our attention on the instant when the weakest bubble density inside the vial visibly began to return to the original state (before shock wave arrival). Looking closely at the circular markings in Figure 2, the above-mentioned density returned to its initial state at 3*T*; thus, the collapse occurred approximately at 2*T*. Because of this, the delay for tandem shock waves generated with 4 kV was chosen to be 2*T* = 66.6 µs. Using the same method, the delays selected for 5 and 6 kV were 99.9 and 166.6 µs, respectively. Obviously, this is rather subjective and depends on the time (*T*) between images. Unavoidably, bubbles of various sizes collapsed and emitted microjets having a variety of diameters. The decimals of the reported delays are meaningless because the methodology used to determine the “optimal” delay had a much higher uncertainty.

### 3.2. Cell Viability

As shown in Table 2, with the settings used in this study, exposure of the conidia to shock waves did not significantly affect cell viability. In most groups, the mean viability was over 80%. No statistically significant difference was observed between them, including the controls (Kruskal–Wallis: *X*^2^ = 11.143, *df* = 17, *p* = 0.849).

### 3.3. Growth and Germination of Conidia after Shock Wave Exposure

Figure 3 shows three representative photographs of a growth sequence of the untreated conidia as observed under the optical microscope. In Figure 3a,b, the size changes are due to swelling or isotropic growth; in Figure 3c, conidia reach the phase of polarized growth (germ tube) and are ready to start the formation of hyphae or somatic growth. The related percentages of germination of conidia are shown afterwards.

All graphs obtained after measuring the swelling, i.e., area growth, of conidia in the 19 groups and the percentage of conidia that formed a germ tube, can be reviewed in the Appendix A). The results corresponding to 200 shock waves are highlighted here (Figure 4). In this case, a different behavior was observed in the treatment groups compared with the control group.

Figure 4a–f show the area versus time graphs corresponding to conidial swelling after each shock wave exposure. All groups, including the treated and control groups, had a similar time trend profile. Initially, the area occupied by the conidia was approximately 20 µm^2^, increasing gradually and reaching a maximum swelling rate between 4 and 8 h. At the end of this period, all conidia had duplicated their area. From this point on, the germ tube phase or somatic growth was initiated in all groups, and the area no longer changed significantly. No area data were obtained for 12 h because of the excessive number of hyphae and agglomerations formed. The percentage of developed germ tubes versus time obtained after exposure to shock waves generated at 4 kV is shown in Figure 4g (single pulse) and Figure 4j (tandem). Stimulation of germ tube formation was observed for almost 80% of the conidia during the first 8 h of incubation. The behavior of the groups that received single-pulse shock waves generated at 5 kV (Figure 4h) and tandem shock waves generated at 5 and 6 kV (Figure 4k,l, respectively) was similar to that of the corresponding control groups, reaching approximately 85% after 12 h. Compared with the control group, which, after 12 h, reached a germination above 80%, single-pulse shock waves generated at 6 kV only achieved approximately 78% (Figure 4i). The statistical analysis also revealed that there was no relationship between the number of conidia that developed a germ tube and the size of the conidia.

### 3.4. Surface Morphology of Aspergillus niger Conidia

Untreated *A. niger* conidia (Figure 5a) have a quasi-spherical morphology with ornaments equinulated on the surface of their cell wall. In contrast, the conidia exposed to single-pulse and tandem shock waves (Figure 5b,c, respectively) had alterations of their cell walls, showing pores with a size ranging between approximately 0.03 and 0.25 µm. Furthermore, the superficial ornaments on the cell walls of all shock-wave-treated conidia were partially worn away, resulting in a smoother surface.

### 3.5. Cell Permeabilization

Figure 6 shows representative confocal laser scanning microscopy images of the middle geometrical plane of conidia selected from three shock-wave-treated groups and the control group. The FITC-FD10 dextran dye (green fluorescence) used to assess membrane permeabilization quantitatively, can be observed both attached to the cell wall and inside shock-wave-exposed conidia. The control group image shows that in the recorded field, only one cell was permeable to this dye. Calcofluor white was used as counterstain because of its affinity to the chitin and polysaccharides, which are components of the conidial cell wall. It is evident that in all treated samples, green fluorescence colocalized with calcofluor white (blue fluorescence) associated with the cell wall stain.

### 3.6. The Effect of Shock Wave Treatments and Incubation Time on the Expression of β-Actin, Wsc4, RhoA and RhoC Genes of Aspergillus niger

Exposure of the conidia to shock waves modified the genetic expression of *A. niger.* In this study, four genes were analyzed: *β-Actin, Wsc4, RhoA* and *RhoC*. These genes were selected because they are involved with cell wall integrity, development of the germ tube and growth of the conidia. Figure 7 shows the results obtained by agarose gel electrophoresis of PCR amplification. The relative quantification of the band intensity values associated with gene expression observed in the electrophoresis gel is shown in Table 3. The expression of *β-Actin* increased over time (from 2 to 8 h). After an incubation time of 2 h, a slight amplification band was observed in the control group (43.5 ng/4 µL). The low-pressure (42.04 MPa) tandem shock wave group had a significant increase in intensity (85 ng/4 µL), indicating an enhanced expression of *β-Actin*. At times 0 and 2 h, no expression of *Wsc4* was detected in the control group. A slight expression (18.5 ng/4 µL) was measured in conidia treated with single-pulse shock waves generated at 4 kV. The expression of *Wsc4* after 8 h was confirmed by the appearance of bands in the control group (54.3 ng/4 µL) and some shock-wave-treated groups. Only the groups treated at a low pressure revealed an amplification band with similar values (59.3 ng/4 µL to 4kV-SP and 57.5 ng/4 µL for 4 kV–T) to the control group. Groups exposed to high-pressure produced either no bands or low-intensity bands. The expression of *RhoA* was enhanced at an incubation time of 8 h among the groups treated with shock waves and compared with the control group. Finally, low expression of *RhoC* could be observed among the groups incubated for 2 h, especially for those exposed to single-pulse shock waves generated at 4 and 5 kV (Lane 4 and 5; 8.5 and 8.3 ng/4 µL, respectively) and tandem shock waves generated at 6 kV (Lane 9). Compared with the other treatments, the intensity of the band increased twice as can be seen for the group treated with tandem shock waves generated at 4 kV (Lane 7). Additionally, it can be observed that *RhoC* was expressed in all of the groups at 8 h.

## 4. Discussion

Direct and indirect effects of shock waves, such as acoustic cavitation, promote cell wall permeabilization [39,40]. Conidia are not naturally permeable to the FITC-FD10 (Stokes–Einstein radius = 2.3 nm) [60]. The presence of this dye inside the cell in shock-wave-treated samples (Figure 6) can be attributed to the transitory permeabilization produced by the effect of the shock waves. Furthermore, green fluorescence attached to the cell wall could probably be due to a disorganization of the components of the cell wall induced by the effects of the shock waves. FD10-FITC was not observed inside or attached to the walls of nontreated cells. This confirms that viable conidia did not capture this dye. Intact cell walls were only stained by calcofluor white. Moreover, nonviable cells revealed a large amount of green fluorescence inside them, but not on their walls. This indicates that the integrity of the structure was not modified after cell death, reaffirming that the presence of FD10-FITC in the cell wall could be attributed to shock wave exposure. The hydrophobic properties of the outer wall of the conidia and the crevices on their surface enhance bubble formation [61]. Both direct and indirect effects of shock waves can cause compression, shearing, and tensile stress. Secondary shock waves may have a scale of tens of micrometers and interact with structures as small as a conidium. Their influence mainly depends on the peak positive and negative pressure, the rise time, the FWHM, and the shock wave rate; however, at the cellular level, it is difficult to determine which shock-wave-induced physical phenomenon is responsible for a specific reaction [62,63,64]. Optimizing other parameters, such as the impulse of the shock wave, i.e., the pressure integrated over time, may be useful to enhance certain desired biological effects.

Because the focal zone of the shock wave source is smaller than the size of the vial, different zones are exposed to different pressure distributions. Nevertheless, after passage of several shock waves, strong acoustic streaming produced inside the transfer pipette promotes all conidia to receive similar pressure variations. Furthermore, bubble growth and collapse as well as secondary shock waves contribute to mixing the solution. As the number of shock waves passing through the suspension increases, more cavitation nuclei are formed. The conidia are damaged by fluid shear and stress, resulting in cell permeabilization. This may be associated with morphological changes on the surface of the conidia (Figure 5). Moreover, due to its short rise time of less than 100 ns, the shock front compresses the conidia increasing the intracellular pressure [64]. This may have biological consequences. It is known that *Aspergillus* sp. conidia have intra- and extracellular protection mechanisms that protect against the increase in external pressure; however, the published data only refer to high-pressure treatments with slow pressure increase rates, not pressure pulses [65,66]. 

Before reaching the conidial suspension, the shock wave passes through the water–vial and vial–suspension interfaces. Because the acoustic impedance of polyethylene (density approximately 0.9 g/cm^3^) is close to that of water, the attenuation of the positive pressure peak was negligible (a maximum attenuation of 0.83% was recorded for shock waves generated at 4 kV), and less bubble formation was observed at the bottom of the vial (the shock wave entrance site) than at the suspension–air interface (exit site). At the fluid–air interface, the wave is inverted after reflection, i.e., it reflects as a rarefaction wave, leading to enhanced cavitation and high strain [62,63].

An advantage of using tandem shock waves, independent of enhancing bubble collapse and increasing the efficiency of genetic transformation [29], is that the same number of shock waves is delivered in half the time as in the single-pulse mode. However, the selection of a convenient delay between shock waves may require a series of preliminary experiments. To optimize the parameters, factorial experimental designs could be useful.

In the study described here, instead of using delays of 66.6, 99.9, and 166.9 µs, we could have chosen 70, 100, and 170 µs as “the best” delays, without obtaining different conclusions. Our decision also depended on the number of frames per second of the images analyzed. Moreover, cavitation is a multibubble phenomenon, and it is uncertain whether microjets and secondary shock waves emitted by the collapse of small bubbles have a stronger influence than those produced after the collapse of large bubbles. The growth and collapse of microbubbles not visible on our high-speed images may have contributed to effects such as the pores seen in Figure 5c. Furthermore, bubbles of different sizes also interact and coalesce.

While the chemistry of fungal cell walls is relatively well understood, little is known about their mechanical properties. Zhao et al. [67,68] used atomic force microscopy to determine the elastic properties of the fungus *Aspergillus nidulans*. Their results revealed that the osmotic conditions in growth medium have a strong effect on cell wall elasticity. The same research group found that the rodlet layer that covers the spore surface of filamentous fungi is significantly softer than the underlying portion of the cell wall. The rodlet layer is easily removed by sonication [67]. A similar effect was observed in our case. Our SEM images (Figure 5) show damage to the cell wall of the conidia and loss of the ornamentation after shock wave exposure.

As mentioned above, during the last decade, shock waves have been reported to be useful for cell permeabilization [23,29,69]. The cell transformation efficiency obtained by using shock waves alone or in synergy with a chemical reactive, such as Lipofectamine, was greater than that obtained with other methods. On the other hand, Peña et al. [56] reported an increase in the content of molecular iodine in cancerous tumor tissue exposed to shock waves in a murine model. Because mammalian cells do not have a protective outer layer, such as the cell wall of fungal conidia, their viability after exposure to shock waves is much lower, reaching a mortality of up to 90% when exposed to 500 shock waves of 18 MPa peak positive pressure [39]. In contrast, the viability, swelling, and germination results obtained in this study indicate that exposure to shock waves in the single-pulse and tandem modes did not generate any severe cellular effects on the conidium. The fungal conidia had a cell viability close to 80% even after being treated with 200 shock waves with a peak pressure of approximately 80 MPa. This may be associated with the presence of a complex protective covering, composed of a polymeric matrix of glucans and chitin, with arrangements of melanin and hydrophobic proteins on the external layer. According to Valsecchi et al. [70], artificially removing the melanin and hydrophobins does not affect germination, since disaggregation of this layer occurs naturally during the swelling phase [48,70]. Our study shows that the conidia were activated and began the process of isotropic growth (swelling) until they reached twice their initial cellular area after treatment with shock waves generated at 4 kV. However, a delay in their growth rate was observed after increasing the pressure from approximately 42 to 66 MPa, although the conidia remained viable compared with the control group, in which the cellular area increased at a constant rate. This delay in the start of swelling could be associated with alterations generated to the fungal cell wall (as shown in Figure 5), albeit without a reduction in viability, since the conidia reached the phase of polarized growth (germ tube formation) in percentages very close to the control. 

Because of their lower elasticity, shock-wave-induced deformation and cell wall permeabilization of conidia may differ from that observed in human cells. Moreover, the repair capacity and the time taken for the repair may also be very different from those of mammalian cells. The cell wall of *A. niger* conidia is a dynamic structure. On the outside, it has a thick hydrophobic layer consisting mainly of glucans, chitin, and associated proteins, which are modified during the diverse growth stages or by stress induced by changes in the environment [7,8,10,11]. Triggering the somatic growth phase after latency requires activation of cellular metabolism (DNA, RNA, and protein synthesis). Conidia are sensitive to external environmental factors, detected through specific proteins associated with the cell wall that are responsible for transmitting the signals related to cell activation, growth, and somatic development of the fungus, under suitable environmental conditions [71]. Two examples are G protein-coupled receptor 1 (Gpr1p), which is associated with the detection of nutrients such as glucose, and the general control nonderepressible 2 (Gcn2p), which is involved in nitrogen sensors [72,73]. In contrast, protein sensors generate and transmit stress signals, activating response pathways, such as the cell wall integrity pathway responsible for the maintenance and biosynthesis of the fungal cell wall. Among these are *Wsc* family genes codifying Wsc proteins (Wscp 1–4). 

The *Wsc4* gene has been reported to have multiple functions in various fungal species, although it has been studied in greater detail in *A. nidulans* and *S. cerevisiae*. Its translated protein (*Wsc4p*) is located in the endoplasmic reticulum (ER) and it functions as an intermediary in the translocation of soluble secretory proteins and in the insertion of membrane proteins in the ER membrane [74]. Furthermore, this protein plays a role in the stress response [75]. In this study, we found that shock waves induced a decrease in the expression of the *Wsc4* gene (Figure 7), which was associated with an inhibition of conidial growth. Interestingly, after exposure to single-pulse shock waves generated at 4 kV, conidial growth was stimulated, and an increase in *Wsc4* gene expression was observed. We believe that a similar expression of *Wsc4* was obtained after treatment with tandem shock waves produced at 4 and 5 kV, probably because at these energy levels, cell damage is relatively low, allowing a faster cell recovery than occurs with cells exposed to shock waves generated at 6 kV. Despite this, the expression of the gene decreased with increased aggressiveness of the treatment, causing a delay in the development of the fungus. Applying single-pulse shock waves generated at 5 and 6 kV resulted in no expression, and with tandem shock waves generated at 5 and 6 kV, diminished expression was observed (Figure 7). These results could be related to the modification of the cell wall structure due to acoustic cavitation and surface abrading, inducing a decrease in downstream signaling and affecting the growth and development of the conidia. A more detailed analysis is required to confirm the presence and modification of these proteins in shock-wave-treated cell walls. Unfortunately, there is no commercially available *Wsc4* antibody against *A. niger*. It must be produced from the purified protein. Nevertheless, our results agree with those obtained by Futagami et al. [76]. They studied the effect of suppressing the *Wsc* genes of *A. nidulans*, reporting changes and defects in the structure of the hyphae in germination and the CWI pathway. Tong et al. [77] reported that suppression of the *Wsc* genes caused a delay in germination and decreased the resistance of conidia.

The *Rho1* gene (a homologous gene of the *RhoA* in *A. niger*) product acts on several downstream effectors involved in controlling cell wall synthesis [78]. These processes have an impact on the subsequent stages of germ tube formation [9], in which the *Rho1* gene product is important in mediating the depolarization and repolarization of the actin cytoskeleton [79] and in the production of glucans in the cell wall by the activation of β-1,3-glucan synthase. In addition, its activation can initiate other molecular processes that include the targeting of secretory vesicles and the activation of mitogen-activated protein kinase (MAPK) in a downstream cascade. This can lead to the modification of the expression of genes related to the biogenesis of the cell wall [80,81]. Our results revealed that in dormant conidia and during the first two hours of growth, the *RhoA* gene was not detected. It was probably not part of the pre-existing gene pool [10,11]. Although its expression was detected when the conidia reached 8 h of incubation, it was observed that its expression in the control groups was lower than in those belonging to the shock-wave-treated groups; however, its effect was not significant on *RhoA* suppression or overexpression. Our results indicate that the application of shock waves on conidia of *A. niger* stimulated the increase in the expression of *RhoC*, a member of the *Rho* family, whose function focuses on cell polarization and vesicle fusion with the plasma membrane [82]. During the first two growth hours in groups where expression of the *RhoC* gene was observed, an increase in the conidia area was also recorded (Figure 6), indicating that *RhoC* could be involved in its development and growth. The phenomena associated with the effects of shock waves and cavitation on the cell wall could activate Rho proteins, acting in turn on the molecular pathways involved in the coupling and fusion of secretory vesicles [83]. The decrease in *RhoC* expression in groups subjected to higher pressure coincided with delays in the isotropic growth 8 h after shock wave exposure. This observation agrees with a study published by An et al. [84], where the suppression of *Rho3* genes in the fungus *Botrytis cinerea* was associated with a reduction in conidia development and diminished colony growth. The special organization of actin in the cytoskeleton of some fungal species is regulated by Rho-family GTPase. The expression of the actin gene in the samples may be related to the conservation of the maintenance of *RhoA* and *RhoC* functions. Kwon and colleagues [9] studied the expression of six Rho GTPases and found that each contributes differently to growth and morphogenesis associated with actin dynamics in *A. niger*.

## 5. Conclusions

A variety of biotechnological applications (cell transformation, secondary metabolite production and stimulation of antibiotic production, among others) have a potential gain from an improved understanding of the effects of shock waves on the germination of *A. niger* conidia. It remains to be determined to what extent the observed alterations caused by shock-wave-induced cavitation and the wear and loss of material from the conidial cell wall were produced by extrinsic factors. This study demonstrates that shock waves modified the cell wall of *A. niger* conidia, inducing variations in the polarized growth and the expression of the four analyzed genes. Shock-wave-induced cavitation did not significantly reduce cell viability. 

The application of shock waves under optimized conditions could be used as an alternative in biotechnology. To the best of our knowledge, this is the first evaluation of ultrastructural changes induced by shock waves on the cell wall and its influence on genetic expression in the early and late stages of conidial germination. In-depth genetic expression studies will be required in the future to establish the detailed modulation of pathways.

## Figures and Tables

**Figure 1 jof-08-01117-f001:**
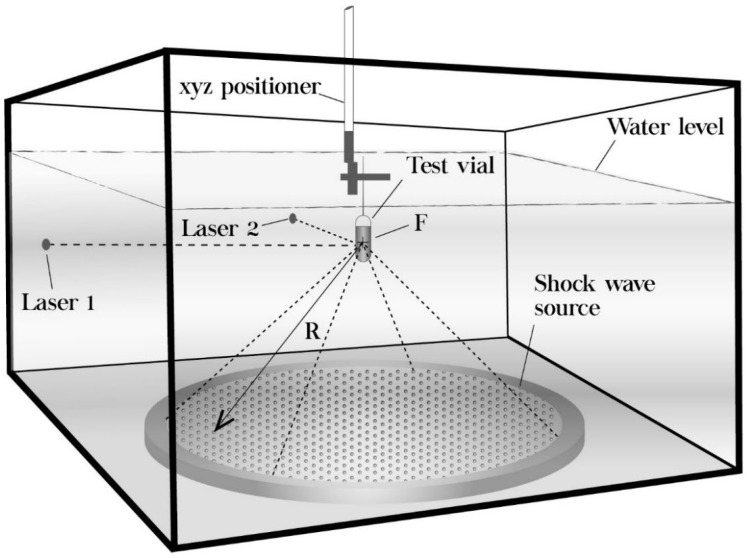
Sketch of the piezoelectric shock wave generator used to expose suspensions of fungal conidia to single-pulse and tandem shock waves. R = 345 mm (Figure adapted from [56]).

**Figure 2 jof-08-01117-f002:**
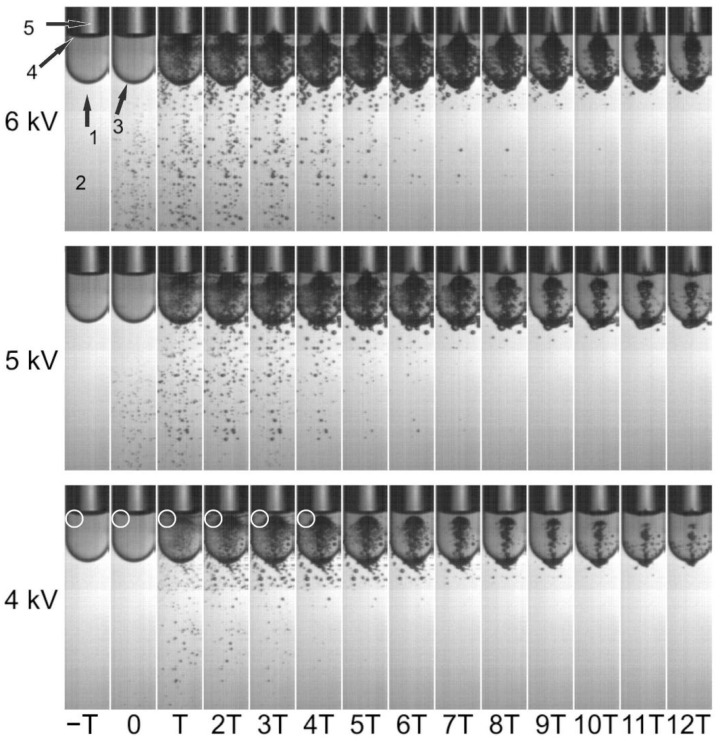
Composition of 14 high-speed images (30,000 fps) showing cavitation bubbles inside a sealed vial containing water and air, after exposure to a single-pulse shock wave, generated at a discharge voltage of 4, 5 and 6 kV. The arrow (1) indicates the direction of the shock wave, which propagated through the water inside the tub (2), penetrated the round-shaped bottom of the vial (3), passed through the water inside the vial and reflected off the water–air interface (4). A diffraction of light (5) from the back side of the vial appears on each image and is irrelevant for the purpose of this study. The first frame where bubbles could be identified was defined as t = 0. In the first six images for 4 kV, circular markings were added to indicate a zone with the smallest cavitation bubbles. (*T* = 1/30,000 s ≈ 33.3 µs).

**Figure 3 jof-08-01117-f003:**
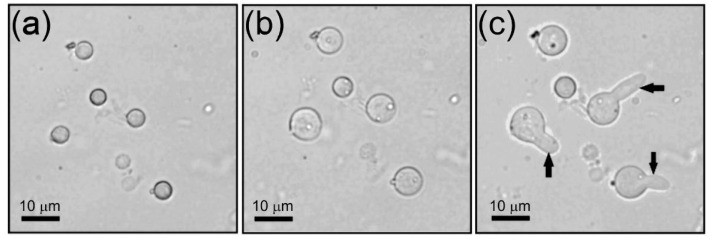
Optical microscope images showing *A. niger* conidia (**a**) at 0 h, i.e., just after being inoculated in the culture medium, (**b**) during swelling or isotropic growth after 6 h, and (**c**) after germ tube formation or polarized growth (see arrows) at 8 h.

**Figure 4 jof-08-01117-f004:**
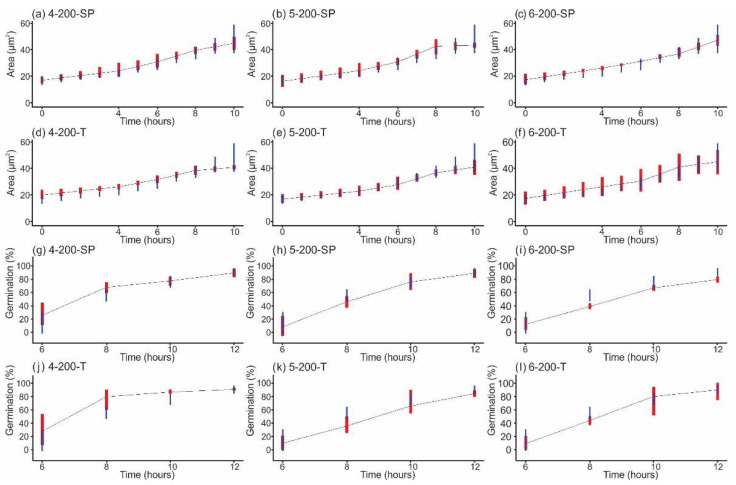
Swelling of conidia versus time after exposure to 200 single-pulse (SP) shock waves generated at (**a**) 4 kV, (**b**) 5 kV, and (**c**) 6 kV, and to 100 tandem (T) events (200 shock waves), generated at (**d**) 4 kV, (**e**) 5 kV, and (**f**) 6 kV. Moreover, it shows the percentage of conidia that formed a germ tube after being exposed to 200 SP shock waves generated at (**g**) 4 kV, (**h**) 5 kV, and (**i**) 6 kV, and to 100 tandem (T) events (200 shock waves) generated at (**j**) 4 kV, (**k**) 5 kV, and (**l**) 6 kV. The graphs resulted from dots corresponding to the average of 100 conidia measured. Vertical red lines represent 95% bootstrap intervals of the growth of shock-wave-treated conidia, and vertical blue lines refer to the 95% bootstrap interval of the control experimental groups. Overlapping intervals indicate a nonsignificant difference.

**Figure 5 jof-08-01117-f005:**
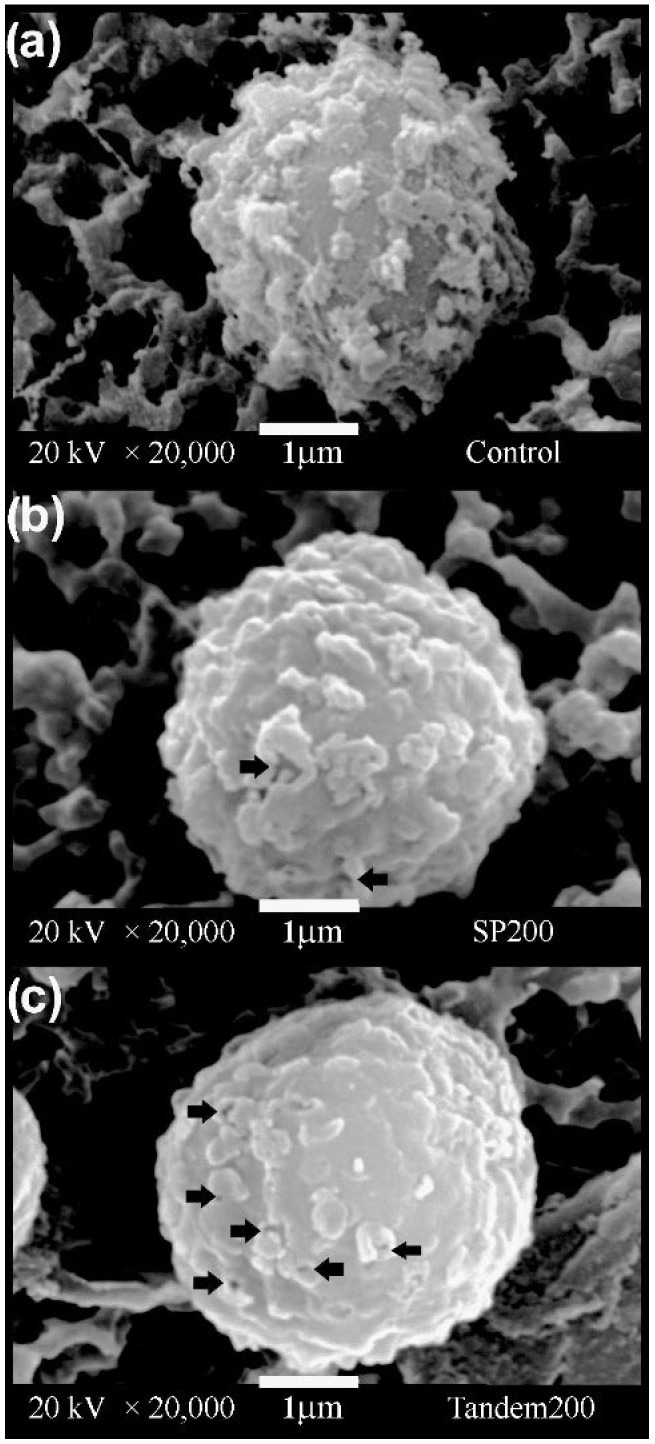
Scanning electron micrographs of *Aspergillus niger* conidia exposed to (**a**) 0 shock waves, (**b**) 200 single-pulse shock waves and (**c**) 100 tandem events. Arrows indicate the presence of pores on the cell wall surface. When compared with the image of a control cell as in (**a**), it is evident that the surface of the shock-wave-treated cells (**b**,**c**) looks smoother.

**Figure 6 jof-08-01117-f006:**
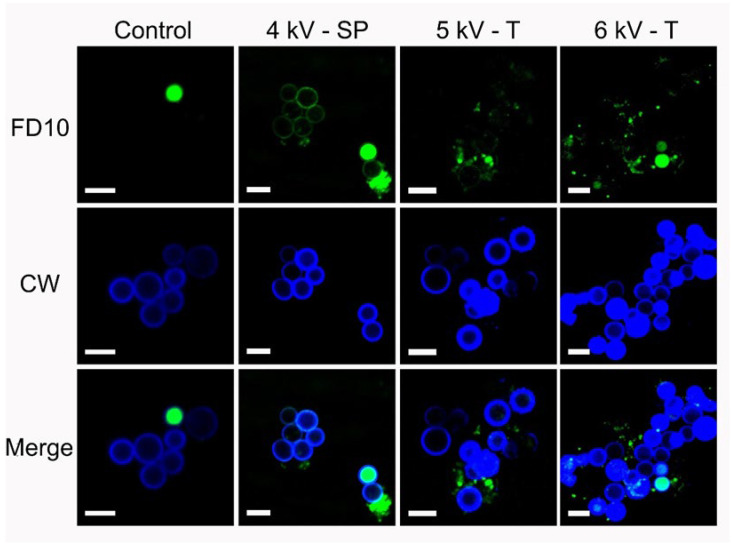
Confocal microscopy images showing the permeabilization of *Aspergillus niger* conidia after exposure to 200 single-pulse shock waves (SP) generated at 4 kV, and 200 tandem shock waves (T) generated at 5 and 6 kV. Representative images of nontreated cells are shown in the first column (Control). FD10 was assessed with merge fluorescence of calcofluor white dye. Scale bar: 5 µm.

**Figure 7 jof-08-01117-f007:**
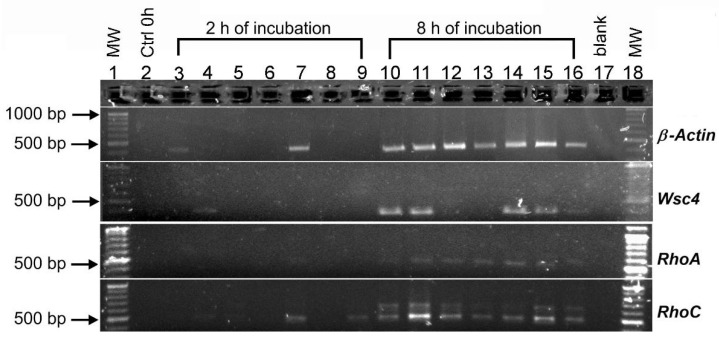
Agarose gel electrophoresis of PCR amplification of *β-Actin*, *Wsc4*, *RhoA* and *RhoC* genes of *Aspergillus niger* conidia after exposure to underwater shock waves and 2 and 8 h of incubation. Lane 1: molecular weight (MW) DNA size marker; Lane 2: control 0 h; Lane 3: control 2 h; Lane 4: 4 kV single-pulse; Lane 5: 5 kV single-pulse; Lane 6: 6 kV single-pulse; Lane 7: 4 kV tandem shock waves; Lane 8: 5 kV tandem shock waves; Lane 9: 6 kV tandem shock waves; Lane 10: control 8 h; Lane 11: 4 kV single-pulse; Lane 12: 5 kV single-pulse; Lane 13: 6 kV single-pulse; Lane 14: 4 kV tandem shock waves; Lane 15: 5 kV tandem shock waves; Lane 16: 6 kV tandem shock waves; Lane 17: negative control; Lane 18: MW.

**Table 1 jof-08-01117-t001:** Primers designed to detect the expression of four genes associated with cell wall integrity (CWI), germ tube development and growth of *Aspergillus niger* conidia.

Genes	Sequence 5′–3′	Product Size (bp)
*β-Actin*	Forward: CACCGGTATCGTTCTGGACTCT	427
	Reverse: ACGGACATCAACATCACACTTCAT	
*Wsc4*	Forward: GCGTGGCTCTTCTCAGATT	378
	Reverse: GCACTGCATCGTTCGCTATC	
*RhoA*	Forward: CGTCCCCTCTCATACCCTGA	506
	Reverse: GCACACATAGTGGAACACGC	
*RhoC*	Forward: GCACGTTTATGCACCCTCAC	518
	Reverse: CCAGAATGAGCGGGGTAGTG	

**Table 2 jof-08-01117-t002:** Average cell viability values for each group.

Single-Pulse	CFU (1 × 10^6^)/mL	Viability (%)	Tandem	CFU (1 × 10^6^)/mL	Viability (%)
Control	1.40 ± 0.17 ^a^	100.00 ± 11.84	Control	1.40 ± 0.17	100.0 ± 11.84
4 ^b^–50 ^c^	1.33 ± 0.23	94.98 ± 8.27	4–50	1.28 ± 0.20	90.9 ± 6.00
4–100	1.39 ± 0.32	98.49 ± 14.28	4–100	1.37 ± 0.37	96.5 ± 18.00
4–200	1.37 ± 0.41	96.40 ± 20.40	4–200	1.31 ± 0.18	93.3 ± 4.33
5–50	1.20 ± 0.17	86.03 ± 4.49	5–50	1.18 ± 0.20	83.7 ± 7.28
5–100	1.16 ± 0.17	82.97 ± 4.57	5–100	1.30 ± 0.26	92.2 ± 10.20
5–200	1.00 ± 0.19	71.54 ± 5.87	5–200	1.18 ± 0.17	84.2 ± 4.80
6–50	1.31 ± 0.19	93.35 ± 4.73	6–50	1.17 ± 0.13	83.7 ± 1.64
6–100	1.25 ± 0.17	89.40 ± 5.24	6–100	1.31 ± 0.12	94.0 ± 0.38
6–200	1.24 ± 0.18	88.75 ± 4.89	6–200	1.22 ± 0.24	86.5 ± 9.60

^a^ Values are means ± standard error of the mean (*n* = 6). ^b^ Discharge voltage (kV) on the shock wave generator. ^c^ Number of applied shock waves.

**Table 3 jof-08-01117-t003:** Relative quantification values obtained by analyzing the band’s intensity in the electrophoresis gel.

		Incubation Time: 2 h (ng/4 µL)						
	Ctrl 0 h	Ctrl	4 kV-SP ^a^	5 kV-SP	6 kV-SP	4 kV-T ^b^	5 kV-T	6 kV-T
*β-Actin*	- ^c^	43.5	-	-	-	85	-	-
*Wsc4*	-	-	18.5	-	-	-	-	-
*RhoA*	-	-	-	-	-	9.2	-	-
*RhoC*	-	-	8.5	8.3	-	15.2	-	11
		**Incubation Time: 8 h (ng/4 µL)**						
	**Ctrl 0 h**	**Ctrl**	**4 kV-SP**	**5 kV-SP**	**6 kV-SP**	**4 kV-T**	**5 kV-T**	**6 kV-T**
*β-A* *ctin*	-	103	120	156	82	131.6	155	90
*Wsc4*	-	54.3	59.3	-	-	57.5	43.3	27
*RhoA*	-	10	14	12	12.3	14.5	14.1	12
*RhoC*	-	16.6	56.7	23	12.6	15.6	26.4	15

^a^ Single pulse shock waves. ^b^ Tandem shock waves. ^c^ Gene expression not detected.

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
