# Peer review of "Effect of Shock Waves on the Growth of Aspergillus niger Conidia: Evaluation of Germination and Preliminary Study on Gene Expression"

_jof, 2022, doi:10.3390/jof8111117_

Round 1

Reviewer 1 Report

The authors have developed a study about the effect of shock waves on Aspergillus niger and their effect on conidia and cell wall changes. I think this article presents interesting results and the methodology is correct but need extra experiment. To evaluate cell permeabilization, the authors need to do a membrane integrity assay, similar case to confirm cell wall modulation by Rho genes described in the manuscripts, it will necessary quantification of cell wall carbohydrates  and evaluate the sensitivity of Aspergillus niger strain to cell wall-stressing agents.

Several comments were listed below.

 1. Figure 4 is poor quality and difficult to differentiate dots from control and wave- treated.

2. I can not see the relationship between actin expression and wave-treatment. In figure 6 the control at 6h has similar expression in 4kV, 5kV and 6 kV in single or tandem wave. Will be useful if the authors could measure the gene expression.

3. Line 424 “Strong expression”, I suggest measuring the expression comparing Lane 7 against to 3.

4.  Figure 6 say RhoA and RhoC, and in the manuscript say Rho1 and 3. Please clarify.

5. Line 456 “ in cell  permeabilization”, again it will necessary membrane integrity assay.

6. Line 543. “we found that shock waves induced a decrease expression…”. This in not necessary that protein of Wsc4 is reduce in amount. Further analysis of Wsc4 protein using antibody will be required.

7. Line 543-458. How the authors could explain Wsc4 has similar expression between at 4kV in single wave, and to 4kV and 5 kV in tandem wave?

8. Line 518 “damage generated to the fungal cell wall”. it will necessary quantification of cell wall carbohydrates, as well as the sensitivity of Aspergillus niger strain to cell wall-stressing agents.

9. Line 551. Which evidence have the authors to say “related to the elimination of Wsc4p from the cell surface”?

Author Response

Your comment:

The authors have developed a study about the effect of shock waves on Aspergillus niger and their effect on conidia and cell wall changes. I think this article presents interesting results and the methodology is correct but need extra experiment. To evaluate cell permeabilization, the authors need to do a membrane integrity assay, similar case to confirm cell wall modulation by Rho genes described in the manuscripts, it will necessary quantification of cell wall carbohydrates  and evaluate the sensitivity of Aspergillus niger strain to cell wall-stressing agents.

Our response:

Thank you for your comments. Your suggestions were addressed as follows:

Your comment:

Figure 4 is poor quality and difficult to differentiate dots from control and wave- treated.

Our response:

To avoid confusion, the graphics were modified, and the quality improved.

Your comment:

I can not see the relationship between actin expression and wave-treatment. In figure 6 the control at 6h has similar expression in 4kV, 5kV and 6 kV in single or tandem wave. Will be useful if the authors could measure the gene expression.

Our response:

Relative quantification of the band´s intensity was performed and included in the revised manuscript to reinforce the results described from the electrophoresis gel observation in the gene expression study.

Your comment:

Line 424 “Strong expression”, I suggest measuring the expression comparing Lane 7 against to 3.

Our response:

The word “strong” was substituted by the values obtained after doing a relative quantification of band intensities.

Your comment:

Figure 6 say RhoA and RhoC, and in the manuscript say Rho1 and 3. Please clarify.

Our response:

The clarification was done. We homogenized the terms “Rho1” and “Rho3” to “RhoA” and “RhoC” throughout the whole revised manuscript.

Your comment:

Line 456 “in cell permeabilization”, again it will necessary membrane integrity assay.

Our response:

A new figure (Figure 6) was included in the revised manuscript showing the qualitative assess of permeabilization of A. niger conidia. This new figure shows fluorescent dye (FITC-FD10 Dextran) internalization in the conidia with calcofluor withe counterstain.

Your comment:

Line 543. “we found that shock waves induced a decrease expression…”. This in not necessary that protein of Wsc4 is reduce in amount. Further analysis of Wsc4 protein using antibody will be required.

Our response:

Unfortunately, there is no commercially available Wsc4 antibody against A. niger. The only available antibody is against Wsc4 of S. cerevisiae; however, to do this assay we would need to test if this antibody exhibits cross-reactivity in A. niger or produce it from the purified protein. Furthermore, standardization would be required. A comment on this was included in the discussion section of the revised manuscript.

Your comment:

Line 543-458. How the authors could explain Wsc4 has similar expression between at 4kV in single wave, and to 4kV and 5 kV in tandem wave?

Our response:

We believe that a similar expression of Wsc4 was obtained after treatment with single-pulse shock waves generated at 4 kV and tandem waves produced at 4 and 5 kV because at this energy levels, cell damage is relatively low, allowing a faster cell recovery than occurs with cells exposed to shock waves generated at 6 kV. A comment on this was included in the discussion section of the revised manuscript.

Your comment:

Line 518 “damage generated to the fungal cell wall”. it will necessary quantification of cell wall carbohydrates, as well as the sensitivity of Aspergillus niger strain to cell wall-stressing agents.

Our response:

We agree that quantification of cell wall carbohydrates and sensitivity analysis to cell-wall stressing agents could result in interesting findings; however, it was not part of the aim of this preliminary study. To determine any damage to the cell wall, we would need to perform a detailed analysis by TEM, which may be the subject of a subsequent study. Quantification of the cell wall carbohydrates may not necessarily be an indication of damage unless it occurs at extensive scale. The fungal cell wall is such a dynamic structure that many changes in carbohydrate composition happen all the time during growth and metabolism. Nevertheless, a comment on this topic was included in the discussion section of the revised manuscript, because your suggestion may be part of a future study. So far, our affirmations are based on the SEM images of shock wave-exposed conidia.

Your comment:

Line 551. Which evidence have the authors to say “related to the elimination of Wsc4p from the cell surface”?

Our response:

The comment was deleted.

Reviewer 2 Report

The manuscript jof-1905579 entitled " Effect of shock waves on the growth of Aspergillus niger conidia: evaluation of germination and preliminary study on gene expression" evaluates the effect of shock waves on the germination of A. niger for potential use in genetic transformation. The topic of the paper fits within the scope of the journal. However, in my opinion the paper is not acceptable for publication in its original form. Major revisions are required in the manuscript. I have some concerns regarding the evaluation of bands intensity/density in RT-PCR. In fact, it is possible to conclude that there are different levels of expression depending on the intensity/density of the band. However, there is a procedure in the software to check the density of the bands. How did the authors do it? Also, in my opinion the authors should review the entire manuscript because it has very long sentences, and sometimes it is very difficult to follow the authors' ideas. In the first three paragraphs of the discussion there is no bibliographic references. Please see my comments below.

Introduction

L85: begin with Aspergillus

Line 121-126: Rewrite this sentence. I found it way too long and difficult to understand.

L168: begin with Aspergillus

Material and Methods

Line 148-149: This experimental setup was performed according to what is described in citation 29? If so, you can say that. If no, please remove this sentence.

L170: What kind of liquid minimal media?

L229: Add reference

L266: and how was it visualized? Please add.

L280-281: How did the authors evaluated the density of the bands? Please clarify.

Results

Line 304-313: This sounds like discussion. Try to avoid citations in results section.

L395: Figures 5b and c, respectively

Discussion

Line 438-470: missing references.

Line 555: Wscp genes?

Line 580-581: Please clarify this sentence and add a reference. It is not clear why you are referring to B. cinerea.

Line 600: I think "exhaustive" may not be the correct word. Please change for "in-depth" ou "comprehensive" for example.

Author Response

Your comment:

The manuscript jof-1905579 entitled " Effect of shock waves on the growth of Aspergillus niger conidia: evaluation of germination and preliminary study on gene expression" evaluates the effect of shock waves on the germination of A. niger for potential use in genetic transformation. The topic of the paper fits within the scope of the journal. However, in my opinion the paper is not acceptable for publication in its original form. Major revisions are required in the manuscript.

Our response:

Thank you for your comments. Your suggestions were followed as described below.

Your comment:

I have some concerns regarding the evaluation of bands intensity/density in RT-PCR. In fact, it is possible to conclude that there are different levels of expression depending on the intensity/density of the band. However, there is a procedure in the software to check the density of the bands. How did the authors do it?

Our response:

The band’s intensity was analyzed by relative quantification using the software coupled to the documentation system Gel Doc Ez Imager (Bio-Rad Laboratories, Inc.) by comparison with the molecular weight marker (Invitrogen SM0331) as a standard. A comment was included in the revised manuscript.

Your comment:

Also, in my opinion the authors should review the entire manuscript because it has very long sentences, and sometimes it is very difficult to follow the authors' ideas.

Our response:

The manuscript was reviewed. Long sentences were shortened and rephrased.

Your comment:

In the first three paragraphs of the discussion there is no bibliographic references.

Our response:

Your suggestion was followed. References are now included in the first three paragraphs of the discussion.

Your comment:

L85: begin with Aspergillus

Our response:

Your suggestion was followed.

Your comment:

Line 121-126: Rewrite this sentence. I found it way too long and difficult to understand.

Our response:

Your suggestion was followed.

Your comment:

L168: begin with Aspergillus

Our response:

Your suggestion was followed.

Your comment:

Line 148-149: This experimental setup was performed according to what is described in citation 29? If so, you can say that. If no, please remove this sentence.

Our response:

Yes. The experimental setup was performed according to what is described in a previous publication [29]. The sentence was rephrased.

Your comment:

L170: What kind of liquid minimal media?

Our response:

We used minimal medium for Aspergillus niger (broth and agar). This is now clearly mentioned in the revised manuscript.

Your comment:

L229: Add reference

Our response:

A brief comment and a reference were added.

Your comment:

L266: and how was it visualized? Please add.

Our response:

The requested information was included (see section 2.9).

Your comment:

L280-281: How did the authors evaluated the density of the bands? Please clarify.

Our response:

The methodology used to evaluate the density of the bands, including information on the software, was added (see section 2.10).

Your comment: 

Line 304-313: This sounds like discussion. Try to avoid citations in results section.

Our response:

We agree. The paragraph was moved to the introduction.

Your comment: 

L395: Figures 5b and c, respectively

Our response:

Your suggestion was followed.

Your comment: 

Line 438-470: missing references.

Our response:

Thank you for the suggestion. Several references were included.

Your comment:  

Line 555: Wscp genes?

Our response:

The sentence was corrected. We were referring to genes, not proteins.

Your comment:  

Line 580-581: Please clarify this sentence and add a reference. It is not clear why you are referring to B. cinerea.

Our response:

The sentence was rephrased, and a reference was included. Our intention is to compare our observations with those reported on the suppression of RhoC associated with the development and growth reduction of B. cinera.

Your comment:  

Line 600: I think "exhaustive" may not be the correct word. Please change for "in-depth" or "comprehensive" for example.

Our response:

Your suggestion was followed.

Reviewer 3 Report

In the manuscript jof-1905579, the authors studied the effects of shock waves on the growth of Aspergillus niger conidia, in particular their germination and gene expression, the purpose of these shock waves being to destabilize cell walls to optimize fungal growth and production of metabolites on an industrial scale.

The following comments/suggestions are offered for the author's consideration:

The subject is really very interesting because the literature is scarce, both on the impact of shock waves and the mechanisms involved on the fungal level, and on the germination of Aspergillus niger which is a less studied filamentous fungus. The development of research and biotechnology in this field could improve the industrial techniques using this fungus.

Although I am not an expert in shock wave techniques, the methods used seem robust to me, as do the phenotypic and gene expression analyzes. The techniques used are varied and specialized. I have just one remark, which will not modify my recommendation: I would have explained the selection of the genes analyzed in this work earlier in the manuscript, in paragraph 2.10, and not in the "Results" section, because on reading the table 1, we ask ourselves the question of the choice of these primers and we only have the answer 5 pages later...

The manuscript is well constructed and well written. References are relevant, methods and results are clearly explained and the discussion is argued and balanced.

For all these reasons, my recommendation is to accept the paper without revision.

Author Response

Your comment:  

In the manuscript jof-1905579, the authors studied the effects of shock waves on the growth of Aspergillus niger conidia, in particular their germination and gene expression, the purpose of these shock waves being to destabilize cell walls to optimize fungal growth and production of metabolites on an industrial scale. The following comments/suggestions are offered for the author's consideration: The subject is really very interesting because the literature is scarce, both on the impact of shock waves and the mechanisms involved on the fungal level, and on the germination of Aspergillus niger which is a less studied filamentous fungus. The development of research and biotechnology in this field could improve the industrial techniques using this fungus. Although I am not an expert in shock wave techniques, the methods used seem robust to me, as do the phenotypic and gene expression analyzes. The techniques used are varied and specialized. I have just one remark, which will not modify my recommendation: I would have explained the selection of the genes analyzed in this work earlier in the manuscript, in paragraph 2.10, and not in the "Results" section, because on reading the table 1, we ask ourselves the question of the choice of these primers and we only have the answer 5 pages later... The manuscript is well constructed and well written. References are relevant, methods and results are clearly explained and the discussion is argued and balanced. For all these reasons, my recommendation is to accept the paper without revision.

Our response:

Thank you for your suggestion. The reason for selecting these specific genes was included in the title of Table 1 of the revised manuscript, emphasizing their relevance regarding our study.

Round 2

Reviewer 1 Report

The authors have developed a study about the effect of shock waves on Aspergillus niger and their effect on conidia and cell wall changes. I think this article has improved with the corrections, extra information in the introduction as well as discussion.

Reviewer 2 Report

The manuscript jof-1905579 entitled "Effect of shock waves on the growth of Aspergillus niger conidia: evaluation of germination and preliminary study on gene expression" was carefully revised in all sections. I can recommend this manuscript for publication.